# Network Inversion
# for
# Training-Like Data Reconstruction

**Pirzada Suhail**[*]
Department of Electrical Engineering
IIT Bombay
Mumbai, IN 400076
psuhail@iitb.ac.in

**Amit Sethi**
Department of Electrical Engineering
IIT Bombay
Mumbai, IN 400076
asethi@iitb.ac.in

## Abstract

Machine Learning models are often trained on proprietary and private data that cannot be shared, though the trained models themselves are distributed openly assuming that sharing model weights is privacy preserving, as training data is not expected to be inferred from the model weights. In this paper, we present Training-Like Data Reconstruction (TLDR), a network inversion-based approach to reconstruct training-like data from trained models. To begin with, we introduce a comprehensive network inversion technique that learns the input space corresponding to different classes in the classifier using a single conditioned generator. While inversion may typically return random and arbitrary input images for a given output label, we modify the inversion process to incentivize the generator to reconstruct training-like data by exploiting key properties of the classifier with respect to the training data along with some prior knowledge about the images. To validate our approach, we conduct empirical evaluations on multiple standard vision classification datasets, thereby highlighting the potential privacy risks involved in sharing machine learning models.

## 1 Introduction

Machine learning models are often trained on proprietary or sensitive data, which cannot be shared openly, yet the trained models themselves are commonly distributed to facilitate various applications assuming that they do not expose the underlying training data. However, recent research suggests that this assumption may not be valid, as it may be possible to infer and reconstruct training or similar data by analyzing the model weights. This potential privacy risk arises from the fact that trained ML models implicitly encode information about the data they were trained on.

Prior research to reconstruct training data has primarily focused on restricted scenarios, such as binary classifiers with fully connected layers trained on a small dataset. In restricted settings, over-parameterized models can easily memorize portions of the training data, leading to successful reconstructions. For under-parameterized models, where there is no possibility of memorization and the models generalize well, reconstructions are typically more difficult. Also in fully connected layers, each input feature is assigned dedicated weights, which may make reconstruction easier as the model captures more direct associations between inputs and outputs. While as in convolutional layers, due to the weight-sharing mechanism, where the same set of weights is applied across different parts of the input, the reconstruction becomes more challenging.

---

[*]Corresponding Author

38th Conference on Neural Information Processing Systems (NeurIPS 2024).

In this paper, we introduce Training-Like Data Reconstruction (TLDR), a novel approach to reconstruct training-like data from vision classifiers with convolutional layers trained on large, complex, and multi-class datasets. At the core of our approach is a network inversion technique that learns the input space corresponding to different classes within a classifier using a single conditioned generator trained to generate a diverse set of samples from the input space with desired labels guided by a combination of losses including cross-entropy, KL Divergence, cosine similarity and feature orthogonality. Inverted samples generated through network inversion are often random, and while inversion may occasionally produce training-like data, our goal is to specifically encourage the generator to reconstruct training-like data. To achieve this, we exploit some key properties of the classifier in relation to its training data.

The classifier is expected to be more confident in its predictions on training samples compared to randomly generated, inverted samples. Mathematically, this can be expressed as:

$$P(y_{\text{in}}|x_{\text{in}}; \theta) \gg P(y_{\text{ood}}|x_{\text{ood}}; \theta)$$

where $P(y|x; \theta)$ represents the softmax output of the classifier for a given input $x$, $\theta$ are the model's parameters, $x_{\text{in}}$ refers to in-distribution data, and $x_{\text{ood}}$ refers to out-of-distribution data.

During training, the model learns to generalize across variations in the training data, making it relatively more robust to perturbations around these samples and the same can be represented by:

$$\frac{\partial f_\theta(x_{\text{in}})}{\partial x_{\text{in}}} \ll \frac{\partial f_\theta(x_{\text{ood}})}{\partial x_{\text{ood}}}$$

Since the classifier has already been optimized on the train set, the gradient of the loss with respect to the weights is expected to be lower for training data compared to random inverted samples, hence:

$$\|\nabla_\theta L(f_\theta(x_{\text{in}}), y_{\text{in}})\| \ll \|\nabla_\theta L(f_\theta(x_{\text{ood}}), y_{\text{ood}})\|$$

where $L$ represents the loss function, $f_\theta(x)$ is the model output for input $x$, and $\nabla_\theta$ is the gradient with respect to the model weights $\theta$. Our main contributions in this paper include:

1. A comprehensive approach to inversion of convolutional vision classifiers using a single conditioned generator.
2. The introduction of soft vector conditioning and intermediate matrix conditioning to encourage diversity in the inversion process.
3. The use of network inversion to reconstruct training-like data by exploiting key properties of the classifier in relation to its training data.

To validate our approach, we conduct extensive inversion and reconstruction experiments on MNIST, FashionMNIST, SVHN, and CIFAR-10, demonstrating that the proposed method is capable of reconstructing training-like data across different domains highlighting the privacy risks.

## 2 Related Works

Network inversion has emerged as a powerful method for exploring and understanding the internal mechanisms of neural networks. By identifying input patterns that closely approximate a given output target, inversion techniques provide a way to visualize the information processing capabilities embedded within the network's learned parameters. These methods reveal important insights into how models represent and manipulate data, offering a pathway to expose the latent structure of neural networks. While inversion techniques primarily began as tools for understanding models, their application to extracting sensitive data has sparked significant concerns. Neural networks inherently store information about the data they are trained on, and this has led to the potential for training data to be reconstructed through inversion attacks. Early works in this space, particularly on over-parameterized models with fully connected networks, demonstrated that it was possible to extract portions of the training data due to the model's tendency to memorize data. This raises significant privacy concerns, especially in cases where models are trained on proprietary or sensitive datasets, such as in healthcare or finance.

Early research on inversion for multi-layer perceptrons in [Kindermann and Linden, 1990], derived from the back-propagation algorithm, demonstrates the utility of this method in applications like digit recognition highlighting that while multi-layer perceptrons exhibit strong generalization capabilities—successfully classifying untrained digits—they often falter in rejecting counterexamples, such as random patterns. Subsequently [Jensen et al., 1999] expanded on this idea by proposing evolutionary inversion procedures for feed-forward networks that stands out for its ability to identify multiple inversion points simultaneously, providing a more comprehensive view of the network's input-output relationships. The paper [Saad and Wunsch, 2007] explores the lack of explanation capability in artificial neural networks (ANNs) and introduces an inversion-based method for rule extraction to calculate the input patterns that correspond to specific output targets, allowing for the generation of hyperplane-based rules that explain the neural network's decision-making process. [Wong, 2017] addresses the problem of inverting deep networks to find inputs that minimize certain output criteria by reformulating network propagation as a constrained optimization problem and solving it using the alternating direction method of multipliers.

Model Inversion attacks in adversarial settings are studied in [Yang et al., 2019], where an attacker aims to infer training data from a model's predictions by training a secondary neural network to perform the inversion, using the adversary's background knowledge to construct an auxiliary dataset, without access to the original training data. The paper [Kumar and Levine, 2020] presents a method for tackling data-driven optimization problems, where the goal is to find inputs that maximize an unknown score function by proposing Model Inversion Networks (MINs), which learn an inverse mapping from scores to inputs, allowing them to scale to high-dimensional input spaces. While [Ansari et al., 2022] introduces an automated method for inversion by focusing on the reliability of inverse solutions by seeking inverse solutions near reliable data points that are sampled from the forward process and used for training the surrogate model. By incorporating predictive uncertainty into the inversion process and minimizing it, this approach achieves higher accuracy and robustness.

The traditional methods for network inversion often rely on gradient descent through a highly non-convex loss landscape, leading to slow and unstable optimization processes. To address these challenges, recent work by [Liu et al., 2022] proposes learning a loss landscape where gradient descent becomes efficient, thus significantly improving the speed and stability of the inversion process. Similarly Suhail [2024] proposes an alternate approach to inversion by encoding the network into a Conjunctive Normal Form (CNF) propositional formula and using SAT solvers and samplers to find satisfying assignments for the constrained CNF formula. While this method, unlike optimization-based approaches, is deterministic and ensures the generation of diverse input samples with desired labels. However, the downside of this approach lies in its computational complexity, which makes it less feasible for large-scale practical applications.

In reconstruction [Haim et al., 2022] studies the extent to which neural networks memorize training data, revealing that in some cases, a significant portion of the training data can be reconstructed from the parameters of a trained neural network classifier. The paper introduces a novel reconstruction method based on the implicit bias of gradient-based training methods and demonstrate that it is generally possible to reconstruct a substantial fraction of the actual training samples from a trained neural network, specifically focusing on binary MLP classifiers. Later [Buzaglo et al., 2023] improve upon these results by showing that training data reconstruction is not only possible in the multi-class setting but that the quality of the reconstructed samples is even higher than in the binary case. Also revealing that using weight decay during training can increase the susceptibility to reconstruction attacks.

The paper [Balle et al., 2022] addresses the issue of whether an informed adversary, who has knowledge of all training data points except one, can successfully reconstruct the missing data point given access to the trained machine learning model. The authors explore this question by introducing concrete reconstruction attacks on convex models like logistic regression with closed-form solutions. For more complex models, such as neural networks, they develop a reconstructor network, which, given the model weights, can recover the target data point. Subsequenlty [Wang et al., 2023] investigates how model gradients can leak sensitive information about training data, posing serious privacy concerns. The authors claim that even without explicitly training the model or memorizing the data, it is possible to fully reconstruct training samples by gradient query at a randomly chosen parameter value. Under mild assumptions, they demonstrate the reconstruction of training data for both shallow and deep neural networks across a variety of activation functions.

In this paper, we explore the intersection of network inversion and training data reconstruction. Our approach to network inversion aims to strike a balance between computational efficiency and the diversity of generated inputs by using a carefully conditioned generator trained to learn the data distribution in the input space of a trained neural network. The conditioning information is encoded into vectors in a concealed manner to enhance the diversity of the generated inputs by avoiding easy shortcut solutions. This diversity is further enhanced through the application of heavy dropout during the generation process, the minimization of cosine similarity and encouragement of orthogonality between a batch of the features of the generated images.

While network inversion may occasionally produce training-like samples, we encourage this process by exploiting key properties of the classifier with respect to its training data. The classifier tends to be more confident in predicting in-distribution training samples than random, out-of-distribution samples, and it exhibits greater robustness to perturbations around the training data. Furthermore, the gradient of the loss with respect to the model's weights is typically lower for training data, which helps guide the generator toward reproducing these samples. Additionally, we incorporate prior knowledge in the form of variational loss to create noise-free images and pixel constraint loss to keep pixel values within the valid range, ensuring the generated images are both semantically and visually aligned with the original training data. By leveraging these insights, we steer the inversion process to reconstruct training-like data and extend prior work on training data reconstruction, which primarily focused on models with fully connected layers, to under-parametrized models with convolutional layers and standard activation functions, trained on larger datasets with regularisation techniques to prevent memorisation.

## 3    Methodology & Implementation

Our approach to Network Inversion and subsequent training data reconstruction uses a carefully conditioned generator that learns diverse data distributions in the input space of the trained classifier.

### 3.1    Classifier

In this paper inversion and reconstruction is performed on a classifier which includes convolution and fully connected layers as appropriate to the classification task. We use standard non-linearity layers like Leaky-ReLU [Xu et al., 2015] and Dropout layers [Srivastava et al., 2014] in the classifier for regularisation purposes to discourage memorisation. The classification network is trained on a particular dataset and then held in evaluation mode for the purpose of inversion and reconstruction.

### 3.2    Generator

The images in the input space of the classifier will be generated by an appropriately conditioned generator. The generator builds up from a latent vector by up-convolution operations to generate the image of the given size. While generators are conventionally conditioned on an embedding learnt of a label for generative modelling tasks, we given its simplicity, observe its ineffectiveness in network inversion and instead propose more intense conditioning mechanism using vectors and matrices.

#### 3.2.1    Label Conditioning

Label Conditioning of a generator is a simple approach to condition the generator on an embedding learnt off of the labels each representative of the separate classes. The conditioning labels are then used in the cross entropy loss function with the outputs of the classifier. While Label Conditioning can be used for inversion, the inverted samples do not seem to have the diversity that is expected of the inversion process due to the simplicity and varying confidence behind the same label.

#### 3.2.2    Vector Conditioning

In order to achieve more diversity in the generated images, the conditioning mechanism of the generator is altered by encoding the label information into an $N$-dimensional vector for an $N$-class classification task. The vectors for this purpose are randomly generated from a normal distribution and then soft-maxed to represent an input conditioning distribution for the generated images. The

argmax index of the soft-maxed vectors now serves as the de facto conditioning label, which can be used in the cross-entropy loss function without being explicitly revealed to the generator.

### 3.2.3 Intermediate Matrix Conditioning

Vector Conditioning allows for a encoding the label information into the vectors using the argmax criteria. This can be further extended into Matrix Conditioning which apparently serves as a better prior in case of generating images and allows for more ways to encode the label information for a better capture of the diversity in the inversion process. In its simplest form we use a Hot Conditioning Matrix in which an $NXN$ dimensional matrix is defined such that all the elements in a given row and column (same index) across the matrix are set to one while the rest all entries are zeroes. The index of the row or column set to 1 now serves as the label for the conditioning purposes. The conditioning matrix is concatenated with the latent vector intermediately after up-sampling it to $NXN$ spatial dimensions, while the generation upto this point remains unconditioned.

### 3.2.4 Vector-Matrix Conditioning

Since the generation is initially unconditioned in Intermediate Matrix Conditioning, we combine both vector and matrix conditioning, in which vectors are used for early conditioning of the generator upto $NXN$ spatial dimensions followed by concatenation of the conditioning matrix for subsequent generation. The argmax index of the vector, which is the same as the row or column index set to high in the matrix, now serves as the conditioning label.

### 3.3 Network Inversion

The main objective of Network Inversion is to generate images that when passed through the classifier will elicit the same label as the generator was conditioned to. Achieving this objective through a straightforward cross-entropy loss between the conditioning label and the classifier's output can lead to mode collapse, where the generator finds shortcuts that undermine diversity. With the classifier trained, the inversion is performed by training the generator to learn the data distribution for different classes in the input space of the classifier as shown schematically in Figure 1 using a combined loss function $\mathcal{L}_{\text{Inv}}$ defined as:

$$\mathcal{L}_{\text{Inv}} = \alpha \cdot \mathcal{L}_{\text{KL}} + \beta \cdot \mathcal{L}_{\text{CE}} + \gamma \cdot \mathcal{L}_{\text{Cosine}} + \delta \cdot \mathcal{L}_{\text{Ortho}}$$

where $\mathcal{L}_{\text{KL}}$ is the KL Divergence loss, $\mathcal{L}_{\text{CE}}$ is the Cross Entropy loss, $\mathcal{L}_{\text{Cosine}}$ is the Cosine Similarity loss, and $\mathcal{L}_{\text{Ortho}}$ is the Feature Orthogonality loss. The hyperparameters $\alpha, \beta, \gamma, \delta$ control the contribution of each individual loss term defined as:

$$\mathcal{L}_{\text{KL}} = D_{\text{KL}}(P\|Q) = \sum_i P(i) \log \frac{P(i)}{Q(i)}$$

$$\mathcal{L}_{\text{CE}} = -\sum_i y_i \log(\hat{y}_i)$$

$$\mathcal{L}_{\text{Cosine}} = \frac{1}{N(N-1)} \sum_{i \neq j} \cos(\theta_{ij})$$

$$\mathcal{L}_{\text{Ortho}} = \frac{1}{N^2} \sum_{i,j} (G_{ij} - \delta_{ij})^2$$

where $D_{\text{KL}}$ represents the KL Divergence between the input distribution $P$ and the output distribution $Q$, $y_i$ is the set encoded label, $\hat{y}_i$ is the predicted label from the classifier, $\cos(\theta_{ij})$ represents the cosine similarity between features of generated images $i$ and $j$, $G_{ij}$ is the element of the Gram matrix, and $\delta_{ij}$ is the Kronecker delta function. $N$ is the number of feature vectors in the batch.

Thus, the combined loss function ensures that the generator matches the input and output distributions using KL Divergence and also generates images with desired labels using Cross Entropy, while maintaining diversity in the generated images through Feature Orthogonality and Cosine Similarity.

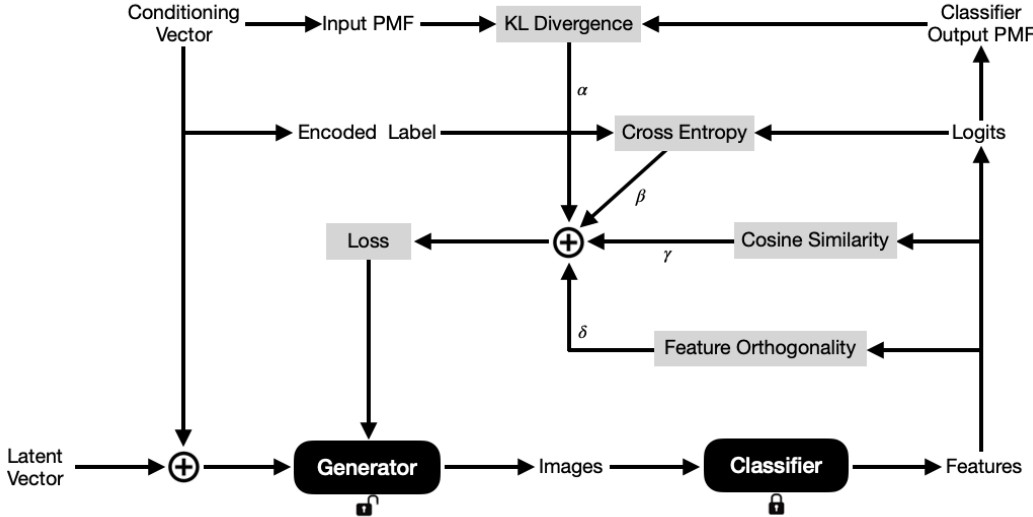

Figure 1: Proposed Approach to Network Inversion

### 3.3.1 Cross Entropy

The key goal of the inversion process is to generate images with the desired labels and the same can be easily achieved using cross entropy loss. In cases where the label information is encoded into the vectors without being explicitly revealed to the generator, the encoded labels can be used in the cross entropy loss with the classifier outputs for the generated images. In contrast to the label conditioning, vector conditioning complicate the training objectives to the extent that the generator does not immediately converge, instead the convergence occurs only when the generator figures out the encoded conditioning mechanism allowing for a better exploration of the input space.

### 3.3.2 KL Divergence

KL Divergence is used to train the generator to learn the data distribution in the input space of the classifier for different conditioning vectors. During training, the KL Divergence loss function measures and minimise the difference between the output distribution of the generated images, as predicted by the classifier, and the conditioning distribution used to generate these images.

### 3.3.3 Cosine Similarity

To enhance the diversity of the generated images, we use cosine similarity to assesses and minimises the angular distance between the features of a batch of generated images across the last fully connected layers, promoting variability in the generated images. The combination of cosine similarity with cross-entropy loss not only ensures that the generated images are classified correctly but also enforces diversity among the images produced for each label.

### 3.3.4 Feature Orthogonality

In addition to the cosine similarity loss, we incorporate feature orthogonality as a regularization term to further enhance the diversity of generated images by minimizing the deviation of the Gram matrix of the features from the identity matrix. By ensuring that the features of generated images are orthogonal, we promote the generation of distinct and non-redundant representations for each conditioning label.

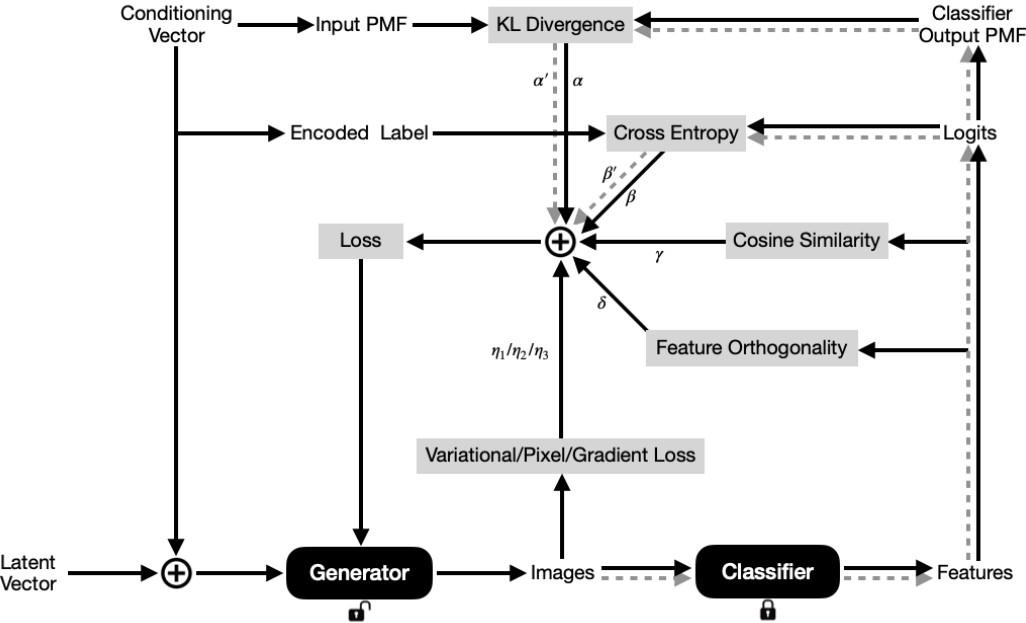

Figure 2: Schematic Approach to Training-Like Data Reconstruction using Network Inversion

## 3.4 Training-Like Data Reconstruction

While Network Inversion enables access to a diverse set of images in the input space of the model for different classes, the inverted samples are completely random. However, Network Inversion can be used for training data reconstruction as shown schematically in Figure 2 by exploiting key properties of the training data in relation to the classifier including model confidence, robustness to perturbations, and gradient behavior along with some prior knowledge about the training data.

In order to take model confidence into account, we use hot conditioning vectors in reconstruction instead of soft conditioning vectors used in inversion, to generate samples that are confidently labeled by the classifier. Since the classifier is expected to handle perturbations around the training data effectively, the perturbed images should retain the same labels and also be confidently classified. Hence, we introduce an $L_\infty$ perturbation to the generated images and pass both the original and perturbed images represented by dashed lines, through the classifier and use them in the loss evaluation. We also introduce a gradient minimization loss to penalise the large gradients of the classifier's output with respect to its weights when processing the generated images ensuring that the generator produces samples that have small gradient norm, a property expected of the training samples. Furthermore, we incorporate prior knowledge through pixel constraint and variational losses to ensure that the generated images have valid pixel values and are noise-free.

Hence the previously defined inversion loss $\mathcal{L}_{\text{Inv}}$ is augmented to include the above aspects into a combined reconstruction loss $\mathcal{L}_{\text{Recon}}$ defined as:

$$\mathcal{L}_{\text{Recon}} = \alpha \cdot \mathcal{L}_{\text{KL}} + \alpha' \cdot \mathcal{L}_{\text{KL}}^{\text{pert}} + \beta \cdot \mathcal{L}_{\text{CE}} + \beta' \cdot \mathcal{L}_{\text{CE}}^{\text{pert}} + \gamma \cdot \mathcal{L}_{\text{Cosine}} + \delta \cdot \mathcal{L}_{\text{Ortho}} + \eta_1 \cdot \mathcal{L}_{\text{Var}} + \eta_2 \cdot \mathcal{L}_{\text{Pix}} + \eta_3 \cdot \mathcal{L}_{\text{Grad}}$$

where $\mathcal{L}_{\text{KL}}^{\text{pert}}$ and $\mathcal{L}_{\text{CE}}^{\text{pert}}$ represent the KL divergence and cross-entropy losses applied on perturbed images, weighted by $\alpha'$ and $\beta'$ respectively while $\mathcal{L}_{\text{Var}}$, $\mathcal{L}_{\text{Pix}}$ and $\mathcal{L}_{\text{Grad}}$ represent the variational loss, Pixel Loss and penalty on gradient norm each weighted by $\eta_1$, $\eta_2$, and $\eta_3$ respectively and defined for an Image $I$ as:

$$\mathcal{L}_{\text{Var}} = \frac{1}{N} \sum_{i=1}^{N} \left( \sum_{h,w} \left( (I_{i,h+1,w} - I_{i,h,w})^2 + (I_{i,h,w+1} - I_{i,h,w})^2 \right) \right)$$

$$\mathcal{L}_{\text{Pix}} = \sum \max(0, -I) + \sum \max(0, I - 1) \qquad \mathcal{L}_{\text{Grad}} = \|\nabla_\theta L(f_\theta(I), y)\|$$

### 3.4.1 Pixel Loss

The Pixel Loss is used to ensure that the generated images have valid pixel values between 0 and 1. Any pixel value that falls outside this range is penalized hence encouraging the generator to produce valid and realistic images.

### 3.4.2 Gradient Loss

The Gradient Loss aims to minimize the gradient of the model's output with respect to its weights for the generated images ensuring that the generated images are closer to the training data, which is expected to have lower gradient magnitudes.

### 3.4.3 Variational Loss

The Variational Loss is designed to promote the generation of noise-free images by minimizing large pixel variations by encouraging smooth transitions between adjacent pixels, effectively reducing high-frequency noise and ensuring that the generated images are visually consistent and realistic.

## 4 Experiments & Results

In this section, we present the experimental results obtained by applying our network inversion and reconstruction technique on the MNIST [Deng, 2012], FashionMNIST [Xiao et al., 2017], SVHN and CIFAR-10 [Krizhevsky et al.] datasets by training a generator to produce images with desired labels. The classifier is initially normally trained on a dataset and then held in evaluation for the purpose of inversion and reconstruction. The images generated by the conditioned generator corresponding to the latent and the conditioning vectors are then passed through the classifier.

The classifier is a simple convolutional neural network with dropout, batch normalization, and leaky-relu activation followed by fully connected layers and softmax for classification. While the generator is based on Vector-Matrix Conditioning in which the class labels are encoded into random softmaxed vectors concatenated with the latent vector followed by transposed convolutions, batch normalization [Ioffe and Szegedy, 2015] and dropout layers [Srivastava et al., 2014] to encourage diversity in the generated images. Once the vectors are upsampled to $NXN$ spatial dimensions they are concatenated with a conditioning matrix for subsequent generation upto the required image size.

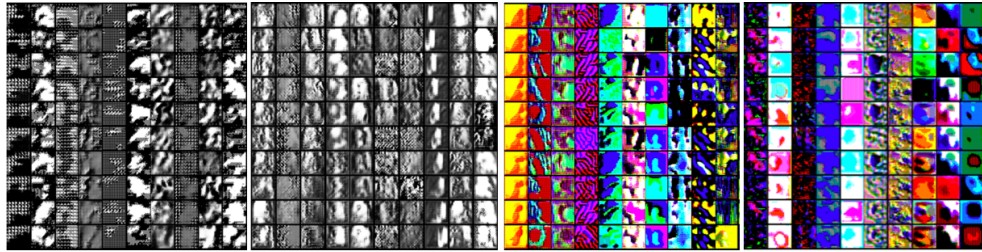

Figure 3: Inverted Images for all 10 classes in MNIST, FashionMNIST, SVHN & CIFAR-10.

The inverted images are visualized to assess the diversity of the generated samples in Figure 3 for MNIST, FashionMNIST, SVHN and CIFAR-10 respectively. While each row corresponds to a different class each column corresponds to a different generator and as can be observed the images within each row represent the diversity of samples generated for that class. It is observed that high weightage to cosine similarity increases both the inter-class and the intra-class diversity in the generated samples of a single generator. These inverted samples that are confidently classified by the generator are unlike anything the model was trained on, and yet happen to be in the input space of different labels highlighting their unsuitability in safety-critical tasks.

The reconstruction experiments were carried out on models trained on datasets of varying size and as a general trend the quality of reconstructed samples degrades with increasing number of training samples. In case of MNIST and FashionMNIST reconstructions performed using three generators each for models trained on datasets of size 1000, 10000 and 60000 are shown in Figure 4.

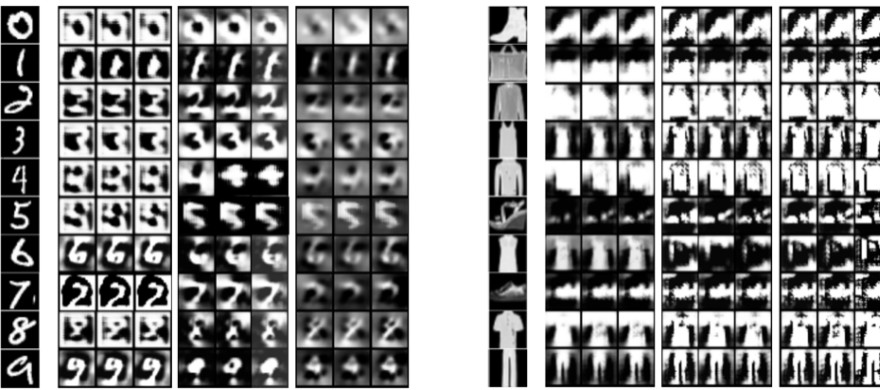

Figure 4: Reconstructed Images for all 10 classes in MNIST and FashionMNIST respectively .

While as for SVHN we held out a cleaner version of the dataset in which every image includes a single digit. In case of CIFAR-10 given the low resolution of the images the reconstructions in some cases are not perfect although they capture the semantic structure behind the images in the class very well. The reconstruction results on SVHN and CIFAR-10 using three different generators on datasets of size 1000, 5000, and 10000 are presented in Figure 5.

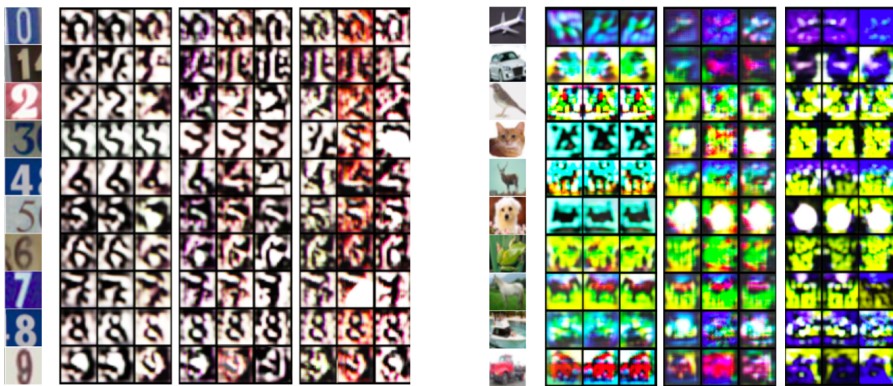

Figure 5: Reconstructed Images for all 10 classes in SVHN and CIFAR-10 respectively.

## 5   Conclusion & Future Work

In this paper, we propose Training-Like Data Reconstruction (TLDR), a novel approach for reconstructing training-like data using Network Inversion from convolutional neural network (CNN) based machine learning models. We begin by introducing a comprehensive network inversion technique using a conditioned generator trained to learn the input space associated with different classes within the classifier using a combination of losses. By exploiting key properties of the classifier in relation to its training data we encouraged the reconstruction of training-like data and demonstrated that machine learning models remain vulnerable to inversion attacks.

As part of the future work, we plan to extend the TLDR approach to more complex architectures to understand privacy vulnerabilities in more advanced neural networks. Further improving the quality of reconstructed samples by leveraging the implicit bias of gradient-based optimization, which tends to memorize a subset of training samples near decision boundaries, will also be explored. Lastly, it would be of interest to evaluate the potential for learning generative models in cooperation with classifiers through network inversion guided by successive weight updates in the classifier during the training process.

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
