# OpenReview forum: "Network Inversion for Training-Like Data Reconstruction"
_NeurIPS.cc/2024/Workshop/SafeGenAi — SafeGenAi Poster_

### Official Review · Reviewer_2KQt · 2024-10-08
**Concerns Regarding Cross-Entropy Focus, Hyperparameter Selection, and Image Diversity in Image Generation Models**

**Rating:** 5
**Confidence:** 5

**Review:**

Comment 1:
In Section 3, the paper need to explore the potential drawbacks of relying heavily on cross-entropy for image generation. For example, focusing solely on classification accuracy may lead to mode collapse, where the generator produces images that are too similar or repetitive. The author should discuss the trade-offs involved in using cross-entropy in this context.

Comment 2:
In Section 3.3, the use of KL divergence to match the input distribution to the output distribution is discussed, but there is no clear explanation for the choice of the hyperparameter α that controls the weight of this term in the overall loss function. Need to show how varying α affects the quality of the reconstructed images and the balance between diversity and accuracy.

Comment 3:
In Section 3.3, the choice of the Gram matrix and the conditions under which orthogonality is enforced are not thoroughly discussed. Furthermore, there is no empirical validation showing that this orthogonality loss significantly improves the diversity of the reconstructed images compared to simpler regularization methods.

Comment 4:
In Figure 3, the paper presents examples of inverted images for different datasets. While these images demonstrate diversity, the qualitative analysis is lacking. The paper should provide quantitative metrics, such as Fréchet Inception Distance (FID) or Inception Score (IS), to evaluate the quality and diversity of the generated images more rigorously.

Comment 5:
In Figure 4, the paper presents reconstructed images for various datasets, but the results for CIFAR-10 are noticeably worse than for MNIST or FashionMNIST. The author briefly mentions the lower resolution of CIFAR-10 as a possible explanation, but this point is not explored in depth.

---

### Official Review · Reviewer_ANXb · 2024-10-09
**This paper presents a novel method for reconstructing training-like data from trained models, highlighting privacy risks effectively.**

**Rating:** 7
**Confidence:** 4

**Review:**

This paper proposes "Training-Like Data Reconstruction (TLDR)," a novel network inversion method to reconstruct training-like data from trained convolutional models, showcasing its efficacy on standard datasets like MNIST and CIFAR-10. The paper is well-written and introduces innovative techniques using a conditioned generator, making complex concepts accessible and highlighting significant privacy risks associated with model sharing. However, the research is limited to benchmark datasets and lacks extensive discussion on scalability and real-world privacy implications. Despite these limitations, the paper significantly contributes to understanding model security vulnerabilities, making it a strong candidate for acceptance.

---

### Official Review · Reviewer_ZWM6 · 2024-10-10
**Limited comparisons with other similar schemes**

**Rating:** 6
**Confidence:** 4

**Review:**

The authors propose a training-like data reconstruction for network inversion and show some experimental results on the inverted data samples. While the motivation is clear, the following comments should be well addressed.

1. The authors should clarify the main difference between the proposed one with other model inversion attacks, and comparisons with such schemes should be provided to evaluate the effectiveness of the proposed one.

2. The paper format seems to be a submission to ICLR, which is inappropriate.